# Carbohydrate Kinases: A Conserved Mechanism Across Differing Folds

**Sumita Roy [1], Mirella Vivoli Vega [2] and Nicholas J. Harmer [1],*** 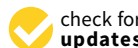

[1]   Living Systems Institute, University of Exeter, Stocker Road, Exeter EX4 4QD, UK; S.Roy@exeter.ac.uk
[2]   Department of Biomedical Experimental and Clinical Sciences, University of Florence, Viale Morgagni 50, 50134 Florence, Italy; mirella.vivoli@unifi.it
*   Correspondence: N.J.Harmer@exeter.ac.uk; Tel.: +44-1392-725179

**Abstract:** Carbohydrate kinases activate a wide variety of monosaccharides by adding a phosphate group, usually from ATP. This modification is fundamental to saccharide utilization, and it is likely a very ancient reaction. Modern organisms contain carbohydrate kinases from at least five main protein families. These range from the highly specialized inositol kinases, to the ribokinases and galactokinases, which belong to families that phosphorylate a wide range of substrates. The carbohydrate kinases utilize a common strategy to drive the reaction between the sugar hydroxyl and the donor phosphate. Each sugar is held in position by a network of hydrogen bonds to the non-reactive hydroxyls (and other functional groups). The reactive hydroxyl is deprotonated, usually by an aspartic acid side chain acting as a catalytic base. The deprotonated hydroxyl then attacks the donor phosphate. The resulting pentacoordinate transition state is stabilized by an adjacent divalent cation, and sometimes by a positively charged protein side chain or the presence of an anion hole. Many carbohydrate kinases are allosterically regulated using a wide variety of strategies, due to their roles at critical control points in carbohydrate metabolism. The evolution of a similar mechanism in several folds highlights the elegance and simplicity of the catalytic scheme.

**Keywords:** phosphorylation; hexokinases; ROK kinases; GHMP kinases; phosphatidylinositol phosphate kinases

## 1. Overview and Importance

Monosaccharides play a critical role in every organism on Earth [1]. They underpin energy generation and central metabolism, and they act as the building blocks for the biosynthesis of polysaccharides. The importance of saccharides and glycans for human health is highlighted by the wide range of glycobiology related genetic disorders [1]; these include over 200 known mutations in glucose-6-phosphate dehydrogenase [2], the most common genetic defect in human enzymes [3]. Many of the glycobiology reactions that cells perform utilize sugars that are phosphorylated at either terminal hydroxyl [1]. Phosphorylation of sugars provides two negative charges that prevent passive diffusion across the cell membrane, enabling intracellular concentration of intermediates [4]. Phosphorylation at both terminal hydroxyls permits reactions that split saccharides, such as the aldolase-catalyzed retro-aldol reactions in glycolysis [5]. Carbohydrate phosphorylation is sufficiently fundamental to cellular function that it is likely to have been developed early in evolution [6]. The reaction is performed by enzymes from five evolutionarily distinct classes (Table 1, Figure 1). These carbohydrate kinases transfer the terminal phosphate from a nucleoside triphosphate (usually ATP) to a free sugar hydroxyl [5]. Defects in carbohydrate phosphorylation have been associated with several diseases; in particular, autosomal galactokinase deficiencies lead to juvenile cataracts within weeks of birth [7,8],

and mutations in glucokinase account for 30–50% of cases of maturity onset diabetes of the young [9]; whilst, galactokinase itself and other carbohydrate kinases are targets for novel therapeutics [10–16].

This review will compare the well-described mechanisms of five classes of sugar kinases (hexokinases, ROK kinases, ribokinases, GHMP kinases, and phosphatidylinositol kinases). Each class has strong experimental evidence for enzyme specificity, regulation, and catalytic mechanism. Despite the structural differences between the enzymes, they show many similarities in how they achieve their functions. This review shows how evolution has achieved the same outcome effectively in several different protein folds.

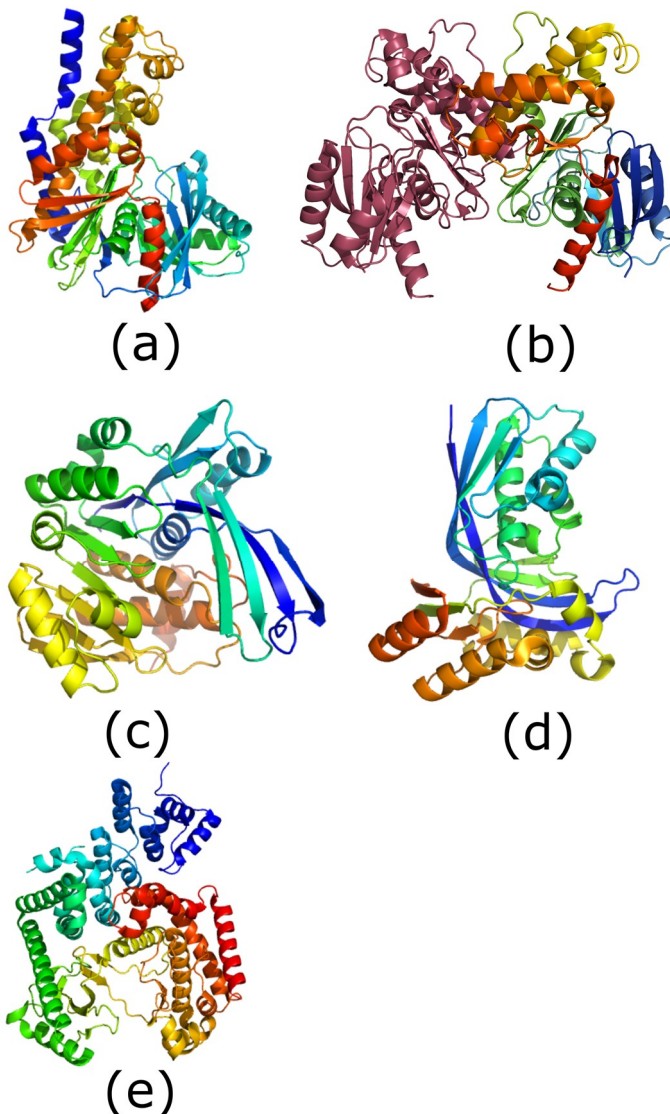

**Figure 1.** Structures of the five carbohydrate kinase classes. All images are shown in rainbow format (blue: *N*-terminus, red: *C*-terminus). (**a**) Structure of human glucokinase (hexokinase class; PDB (protein data bank) ID: 4IWV). (**b**) Structure of *Bacillus subtilis* fructokinase dimer: second molecule shown in raspberry (ROK kinase class; PDB ID: 1XC3 [17]). (**c**) Structure of *Escherichia coli* ribokinase (ribokinase class; PDB ID: 1RKD; [18]). (**d**) Structure of *Aquifex aeolicus* IspE (GHMP kinase class; PDB ID: 2V2Z; [19]). (**e**) Structure of human PIK3C3 (phosphatidylinositol phosphate kinase class; PDB ID: 3IHY). Images generated using PyMOL (version 1.8, Schrodinger, New York, NY, USA).

**Table 1.** Overview of the carbohydrate kinase families.

| Carbohydrate Kinase Family | Common Substrates | Native Phosphate Donors (Minor Donors in Parentheses) | Pfam ID | References |
|---|---|---|---|---|
| Hexokinase | Glucose, mannose, fructose | ATP (ITP) | PF00349, PF03727, PF02685 | [20–24] |
| ROK Kinase | Glucose, allose, fructose, *N*-acetylglucosamine, *N*-acetylmannosamine | ATP (polyphosphate) | PF00480 | [17,25,26] |
| Ribokinase | Ribose, 2-deoxy-D-ribose, adenosine | ATP, ADP (GTP, ribonucleotide) | PF00294 | [27–32] |
| GHMP Kinase | Galactose, *N*-acetylgalactosamine, | ATP (GTP, ITP) | PF00288 | [8,33–37] |
| Phosphatidylinositol kinase | Phosphatidylinositol, phosphatidylinositol phosphates | ATP (GTP) | PF00454, PF01504 | [38] |

## 2. Hexokinases

### 2.1. Overview of the Family

The hexokinases (HK) are perhaps the most familiar carbohydrate kinases. They typically phosphorylate sugars at the hydroxyl at the opposite terminus to the anomeric carbon (i.e., the 6′-position in a six carbon sugar). These enzymes phosphorylate a wide range of sugars and they mostly prefer ATP as the phosphate donor [39]. Hexokinases fall into two main classes, traditionally labelled hexokinase and glucokinase (GK), following the nomenclature of the human isoforms [40]. Eukaryotes generally have multiple HK paralogues and a single GK: the preferred substrate of all these enzymes is glucose [20]. By contrast, most prokaryotic hexokinases belong to the GK family and have a wide range of preferred substrates. These substrates include glucose, ribulose, gluconate, xylulose, glycerokinase, fructose, rhamnose, and fucose [4,20]. Glucokinase activators have been tested in the clinic as anti-diabetics [14,41,42]; whilst, hexokinase inhibitors have been proposed as therapeutics targeting eukaryotic parasites that are reliant on glycolysis [11,15].

### 2.2. Hexokinase Specificity

Hexokinases generally show a moderate specificity for their target sugars, although some are more promiscuous [43]. Higher eukaryotes, with several hexokinase paralogues, are generally slightly less specific, phosphorylating mannose and fructose at the 6′-hydroxyl, in addition to glucose [20]. Lower eukarya and bacteria, in contrast, generally have separate enzymes for each substrate, with higher specificity [20], although some bispecific enzymes have been characterized [20,44]. This pattern is reversed when considering the phosphate donor specificity. Most eukaryotic hexokinases show a strong preference for ATP as the phosphate donor (e.g., rat hexokinase has 24-fold lower $K_M$ for ATP when compared to inosine triphosphate, ITP, the next most preferred) [23]. Bacterial hexokinases are more competent to substitute ATP for ITP or ADP (adenosine diphosphate) [21,22,24].

### 2.3. Allosteric Regulation of Hexokinases

Many HKs act at important regulation points for cellular biochemical pathways. Consequently, many are regulated allosterically by feedback mechanisms to ensure appropriate activity. Eukaryotic hexokinase isoforms I and III contain a second, vestigial HK domain that acts as a binding site for allosteric effectors [20]. Binding of the product glucose-6-phosphate ($K_i$ ranging from 0.06–3.4 mM) to this site causes a conformational change that inhibits enzyme activity [45]. This can be alleviated by binding of the activators phosphate or citrate to the same site [20]. The hexokinase II isoform lacks this regulatory domain. Many cancers that overuse glycolysis (the "Warburg effect") therefore significantly overexpress HK II to drive glycolysis, highlighting the importance of this regulation [46,47]. Yeast HK uses an alternative allosteric regulation, where ATP binds between two HK protomers to generate a non-symmetric dimer. This dimer shows half-site reactivity, and so the effective enzyme

rate is reduced to half [48]. GKs, in contrast, are allosterically regulated via GK regulatory proteins (GKRP), which bind in the absence of glucose [49,50]. GK inhibition by GKRPs is enhanced by the binding of fructose-6-phosphate (a downstream product of GK activity), but alleviated by fructose-1-phosphate [12,51]. The allosteric regulation of these enzymes makes them an attractive target for therapy.

### 2.4. Structure and Catalytic Mechanism of Hexokinases

A wide range of structures of hexokinases from both eukaryotes and prokaryotes has been determined using X-ray crystallography (e.g., [22,52,53]). The hexokinase fold consists of a large $\alpha/\beta$ domain and a small $\alpha/\beta$ domain, separated by a deep cleft that contains the active site (Figure 2a). On binding to the sugar substrate to the bottom of the interdomain cleft, there is a rigid body closure of the domains [54]. A shear movement of the $\alpha$3 helix and core $\beta$-sheet causes four loops of the large domain to close over the sugar substrate to complete the active site, with two of these (L1, L4) being highly conserved [45,54]. The induced fit of the enzyme over the sugar is essential to form the functional nucleotide binding site [45,54]. This site is a deep cleft that is formed between the two domains after sugar binding. Despite this, hexokinases are generally agreed to show a random sequential mechanism, where either substrate may bind first [55]. Five conserved motifs bind to the nucleotide, including residues from both subunits [54], which also coordinate an $Mg^{2+}$ ion binding to the nucleotide phosphates.

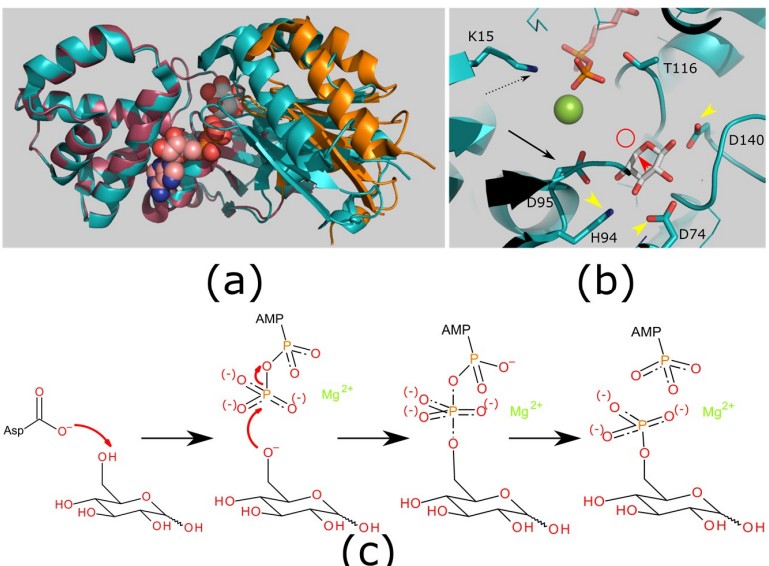

**Figure 2.** Structure and mechanism of hexokinases. (**a**) Hexokinases show a conformational change on binding substrate to create the ATP binding site. Structures of an apo enzyme (raspberry/orange; PDB ID: 2E2N) and ligand bound enzyme (teal; PDB ID: 2E2Q [43]). Ligands xylose (white carbons; substrate analog) and ADP (carbons pink; product). The protein structures have the large domain (left: teal/raspberry) superimposed to highlight the movement in the small domain (teal/orange). There is a substantial rigid body closing of the domains to enclose the substrate. (**b**) Catalytic mechanism. The reactive hydroxyl (would be located at the red circle, attached to an additional carbon at the point indicated by the red arrowhead) is deprotonated by an aspartic acid (catalytic base; black arrow; D95). This will attack the -phosphate of ATP. The pentacoordinate transition state is stabilized by the metal ion (green sphere) and a positively charged side chain (black dashed arrow; K15). Additional side chains hold the sugar in position (yellow arrowheads; D74/H94/D140). (**c**) Generic reaction mechanism for carbohydrate kinases (hexokinase shown). The reaction is initiated by a catalytic base abstracting a proton from the reactive hydroxyl (left). The oxygen atom then attacks the -phosphate of ATP (second left), forming a pentacoordinate transition state (second right). This is stabilized by a divalent cation, and by the protein (not shown). This transition state resolves leaving ADP and the phosphorylated carbohydrate (right). Images generated using PyMOL version 1.8 (**a**,**b**) and BIOVIA Draw (version 2017 R2, 3DS, Paris, France) (**c**).

The mechanism of catalysis is predominantly a mixture of base catalysis, metal ion catalysis, and the formation of a near-attack conformation (Figure 2b,c). The 6′ hydroxyl is deprotonated by a conserved aspartic acid, turning the 6′ O$^-$ into a stronger nucleophile. This then attacks the γ-phosphate of the nucleotide, which is presented by the enzyme in an ideal location for reaction. The Mg$^{2+}$ ion and a conserved positively charged side chain [43] stabilize the negative charge of the ATP and especially the transition state. In human glucokinase, a lysine side chain also acts as a general acid to protonate the leaving phosphate [56]. As the β-phosphate interacts only with small domain, it is expected that, after catalysis, the ADP product will be removed from the active site by an outward movement of the small domain, releasing the sugar-phosphate product [45,54].

## 3. ROK Kinases

### 3.1. Overview of the Family

The ROK (repressor, open reading frame, kinase) kinases are predominantly bacterial enzymes that are structurally related to the HKs [57–59]. These enzymes are generally used to phosphorylate monosaccharides that are produced from the catabolism of cellular components or complex polysaccharides that are used as alternative sources of carbon and energy. Known substrates include glucose [25], allose, fructose [17], *N*-acetylglucosamine [26], and *N*-acetylmannosamine [60]. *N*-acetylglucosamine is of particular relevance for recycling of the peptidoglycan cell wall [26,57] and the utilization of chitin. They have a particular importance for free-living bacteria that require flexibility in their food sources.

### 3.2. ROK Kinase Specificity and Regulation

Few ROK kinases have been extensively characterized for their substrate specificity. Those that have been studied are selective for a narrow range of related sugars, with little activity against other sugars [25,57]. Consistent with this, there is good evidence of sequence motifs for binding different substrates [61]. Most ROK kinases characteristically strongly prefer ATP as a phosphate donor, and require a magnesium ion in a similar manner to the HKs [58]. However, some of these enzymes can additionally utilize polyphosphate as a phosphate donor [25]. ADP, in contrast, acts as a product inhibitor [62]. Archaeal ROK hexokinases and glucokinases show little evidence of allosteric effects from glycolytic intermediates at physiological concentrations [62]. In contrast, the ROK glucokinase from the antibiotic producing bacterium *Streptomyces coelicolor* is allosterically regulated (both activated and inhibited) by a variety of five-carbon sugars [63].

### 3.3. ROK Kinase Structure and Catalytic Mechanism

The ROK kinases have a related structure to the HKs, with a large α/β domain and a small α/β domain (Figures 1b and 3a). Unlike the HKs, ROK kinases all form dimers: some members form a tetramer (dimer of dimers) [25]. Two sequence motifs ("ROK motif" (CXCGXXGC) and ATP binding motif (DXGXT)) are found in the large domain [58,59]. The ROK motif acts as a zinc finger motif and all ROK kinases include structural zinc. As with the HKs, the carbohydrate binding site is formed at the cleft between the two domains, with the carbohydrate held in position by amino acid side chains from both domains [59,64]. Again, in common with the HKs, the binding of the carbohydrate causes a re-arrangement of the two domains, and the ATP binding site forms only after the carbohydrate binds. Kinetic analysis indicates that the enzymes show an ordered Bi–Bi sequential mechanism where the carbohydrate binds first, followed by ATP [64], unlike the hexokinases. A divalent cation is again required for catalysis: characterized ROK kinases can utilize magnesium or manganese [62]. Similarly to HKs, ROK kinases catalyze the phosphorylation of carbohydrates using a conserved aspartic acid to act as a general base and deprotonate the reactive hydroxyl group [17,64]. The negatively charged oxygen atom then attacks the ATP γ-phosphate, forming the pentacoordinate transition state. This is

stabilized by the divalent cation, and possibly by positively charged residues located near to the γ-phosphate [17].

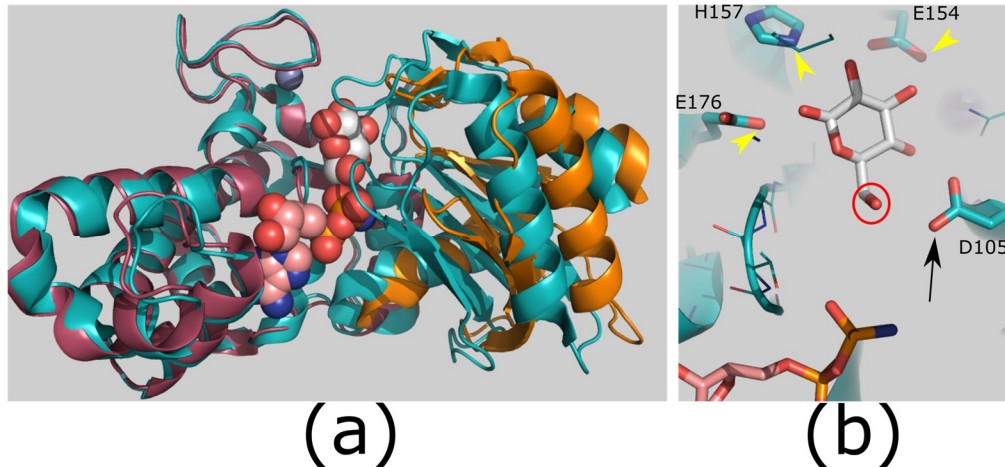

**Figure 3.** Structure and mechanism of ROK kinases. (**a**) ROK kinases show a conformational change on binding substrate to create the ATP binding site. Structures of a ROK glucokinase apo enzyme (raspberry/orange; PDB ID: 3VGK) and ligand bound enzyme (teal; PDB ID: 3VGL [64]). Ligands glucose (white carbons; substrate) and adenylyl-imidodiphosphate (AMP-PNP; carbons pink; substrate analog); zinc shown as gray sphere. The protein structures have the large domain (left: teal/raspberry) superimposed to highlight the movement in the small domain (teal/orange). There is a substantial rigid body closing of the domains to enclose the substrate. (**b**) Catalytic mechanism. The reactive hydroxyl (red circle) is deprotonated by an aspartic acid (catalytic base; black arrow; D105). This will attack the γ-phosphate of ATP. Additional side chains hold the sugar in position (yellow arrowheads; E154/H157/E176). Images generated using PyMOL version 1.8.

## 4. The Ribokinase Family

### 4.1. Overview of the Family

The ribokinase (RK) family of carbohydrate kinases includes adenosine kinases (AK; this non-carbohydrate kinase will not be discussed further), fructokinases and phosphofructokinases, inosine-guanosine kinase, 2-dehydro-3-deoxygluconokinase, and 6-phosphotagatokinases [29]. Enzymes in this family catalyze the phosphorylation of a variety of carbohydrates, nucleosides and cofactor precursors. Most of these enzymes have broad importance to cells in anabolic reactions to synthesize amino acids, nucleotides, and thiamine diphosphate [18,65]. An exception is the phosphofructokinases (PFKs), which are core enzymes in glycolysis [66]. The PFKs are amongst the best examples of enzyme allosteric regulation.

### 4.2. Ribokinase Specificity

Most RKs have high specificity towards their substrates, in comparison to similar naturally occurring molecules. For example, whilst ribokinases will phosphorylate 2-deoxy-D-ribose with only moderately raised $K_M$ in comparison to D-ribose, very low activity is observed with any of the other D-pentoses or D-fructose [29]. One family of RKs from the archaeal order Methanococcales contain an adaptation, allowing them to accept both glucose and fructose-6-phosphate as substrates. These enzymes perform two steps in the classical Embden–Meyerhof–Parnas glycolytic pathway [44,67]. RKs are also considerably more promiscuous in their substrate specificity for the phosphate donor than HKs. *E. coli* RK has activity with GTP as a phosphate donor as well as the preferred substrate ATP [28]; whilst, *Salmonella typhimurium* RK will accept several ribonucleotides and deoxyribonucleotides [30]. Other members of the family (e.g., *Trypanosoma brucei* adenylate kinase) also accept alternative nucleotides with $K_M$ in the same order of magnitude as ATP [32]. Some unusual ribokinases use

alternative phosphate donors. Archaeal RKs have been characterized that prefer donors including ADP and acetyl-phosphate [31]; whilst, an amoebal PFK uses inorganic pyrophosphate as the donor [68]. It is likely that other enzymes in this class have unusual or multiple donors, but these are yet to be tested for.

*4.3. Allosteric Regulation of Ribokinases*

Similarly to the HKs, there is strong evidence that some RKs are allosterically regulated, again reflecting their critical roles in metabolic pathways. Phosphofructokinases are amongst the best understood allosterically regulated proteins. Both prokaryotic [66,69] and eukaryotic [70] PFKs show allosteric regulation to feedback through glycolysis. Indeed, eukaryotic PFKs have over 20 known effectors and two effector binding sites [70]. The *E. coli* phosphofructokinase-2 shows elegant allosteric regulation by Mg-ATP (as ATP is both a substrate and the end product of glycolysis). A second Mg-ATP molecule binds close to the active site, slowing the reaction without competing with the carbohydrate substrate [71,72]. RKs are allosterically regulated by a range of phosphorylated products [73–75]. These compete for a binding site, with the degree of charge on the phosphorus atom directing either activation or inhibition [29]. Several RKs also require a potassium ion in a structural role that is close to the nucleotide binding pocket [76–79]. These mechanisms are very different from those that the HKs use for control. This reflects that the RK fold offers different opportunities for regulation compared to the more dynamic HKs.

*4.4. Structure and Catalytic Mechanism of Ribokinases*

Similarly to HKs, RKs are two-domain proteins, with a large domain and a small domain [80]. The large domain consists of a large $\beta$-sheet flanked on both sides of ten $\alpha$-helices, and contains the majority of the substrate (carbohydrate and nucleotide) binding sites. The second domain is smaller in comparison to the large domain than in the HKs. It consists of a core of four $\beta$-strands that act as a lid over the active site (Figure 4a) [18]. Similarly to the HKs, RKs can be monomeric or dimeric. Ribokinases are generally dimeric, with the protruding $\beta$-sheets of the small domains forming an extended $\beta$-clasp that forms the dimer [18]. In contrast, adenosine kinases are monomeric, with an insert into the small domain blocking the dimerization site [29]. The carbohydrate binding site lies between the large and small domains, although mostly this is contained in the large domain. The nucleotide binding site lies in a groove on the large domain adjacent to the carbohydrate binding sites [18,80]. Unlike the HKs, the two binding sites are present in the apo-protein. However, there is a conformational change between the two domains on carbohydrate binding, with the small domain rotating 15–30° relative to the large domain to cover the binding site [29,80]. This is a driven by a series of hydrophobic interactions between the sugar and the lid. This change is not essential to form the nucleotide binding site [18], but it does increase the enzyme's affinity for ATP. In RK, the $\gamma$-phosphate group of ATP is brought close to the O5′ atom of the ribose at the edge of the lid, close to the optimal distance for reaction [76]. The conformational change also affects the catalytic site of the enzyme, and so it is again required for catalysis. This suggests that RKs will again use an ordered sequential Bi Bi mechanism [18,29]. RKs also bind a divalent cation, which is essential for function. Both eukaryotic and prokaryotic ribokinases show a complete loss of catalytic activity without these cations. Magnesium is preferred for *E. coli* ribokinase, but a range of other divalent cations can play the same role as $Mg^{2+}$ in the enzymatic reaction [29,76,78], such as $Mn^{2+}$, $Co^{2+}$, $Ca^{2+}$, $Ni^{2+}$, and $Cu^{2+}$ [28]. Some RKs also require a monovalent cation (usually potassium, but also cesium) to organize the ATP loops [76,78]. This shapes the nucleotide binding pocket, enhancing both the substrate binding affinity and the catalytic activity of ribokinases [78]. Additionally, in crystallographic structures, a second ion that is bound to the phosphates of the nucleotide has been observed ([81,82]). The catalytic relevance of this remains unclear.

The ribokinase mechanism follows the patterns that were observed for other carbohydrate kinases. The ternary complex of enzyme, carbohydrate, and ATP creates the ideal microenvironment for the

phosphoryl transfer. The most strongly conserved residue is an aspartic acid [29,78] (Asp255 in *E. coli* ribokinase) that functions as a catalytic base to abstract a proton from the reactive hydroxyl group [18], as is seen with other carbohydrate kinases (Figures 2c and 4b). This now negatively charged group makes a nucleophilic attack on the γ-phosphate group of ATP. Allosteric activators draw the β-phosphate away from the γ-phosphate, weakening the bonds between them and facilitating catalysis [29]. The pentacoordinate transition state is stabilized by an anion hole that is formed by the main chain amides from the N-terminal end of α8 [18], and the sidechain of a conserved lysine residue (Lys43 in *E. coli* ribokinase) that is also implicated in sugar binding (Figure 4b) [18,78]. The reaction mechanism for those RKs with good biochemical data is an ordered sequential Bi–Bi mechanism, where the sugar substrate binds first, followed by ATP [29]. This is consistent with the structural observations that carbohydrate binding helps to facilitate ATP binding.

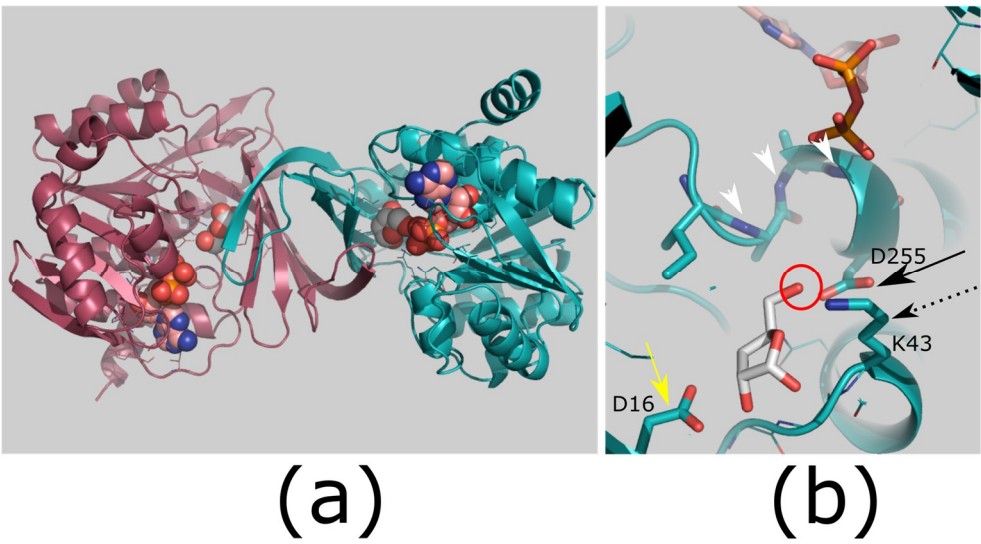

**Figure 4.** Structure and mechanism of ribokinases. (**a**) Dimerization of ribokinases. The small β-sheet domains of two monomers (raspberry and teal) join to form a β-clasp. (**b**) Proposed mechanism for ribokinases. The reactive hydroxyl (red circle) is deprotonated by an aspartic acid (catalytic base; black arrow; D255). This will attack the γ-phosphate of ATP. The pentacoordinate transition state is stabilized by a lysine side chain (black dashed arrow; K43) and by an anion hole produced by main chain amides (white arrowheads). An additional side chain helps to bind the sugar hydroxyls (yellow arrow; D16). A divalent cation was not observed in this crystal structure but is essential for catalysis. Colors: ligand bound enzyme (teal; PDB ID: 1RKD [18]). Ligands ribose (white carbons; substrate) and ADP (carbons pink; product). Images generated using PyMOL version 1.8.

## 5. The GHMP Kinase Family

### 5.1. Overview of the Family

A fourth group of carbohydrate kinases belong to the GHMP kinase family (standing for galactokinase, homoserine kinase, mevalonate kinase, and phosphomevalonate kinase) [83]. GHMP kinases belong to a wider structural fold that includes diverse proteins. These include several carbohydrate kinases: in addition to the galactokinase for which the family is named, these include *N*-acetylgalactosamine (GalNAc) kinase [84], galacturonic acid kinase [85], arabinose kinase [86], and one group of heptose-7-phosphate kinases [87]. Human galactokinase is important for galactose catabolism (via the Leloir pathway), and it has been suggested as a target for drug interventions to suppress galactosemia (caused by an accumulation of the product galactose-1-phosphate when the next Leloir pathway enzyme is absent) [10,88–90]; or, for use in enzyme replacement therapy in the (rarer) galactosemias that is caused by a defect in galactokinase [91].

## 5.2. GHMP Kinase Specificity

Most characterized GHMP carbohydrate kinases have a reasonably tight specificity, targeting their preferred substrate and very similar molecules. The best characterized are the galactokinases. Human galactokinase (GALK1) is highly specific for its sugar substrates phosphorylating only α-D-galactose and closely related analogues, galactosamine, and 2-deoxy-galactose [8]. In contrast, human *N*-acetylgalactosamine kinase (GALK2) has broader specificity, and it will phosphorylate both *N*-acetylgalactosamine and other sugars that share the 2′-acetylamine modification [92,93]. Prokaryotic galactokinases have more relaxed substrate specificity, accepting a variety of galactose analogues as substrates [33,94–97]. Far fewer data are available on the nucleotide specificity of the GHMP carbohydrate kinases. The available data suggest that they have a strong preference for ATP. One case of a kinase that can also use GTP and ITP with lower activity has been reported [33]. As with other carbohydrate kinases, a magnesium ion is also required for activity [87,98].

## 5.3. Structure and Catalytic Mechanism of GHMP Carbohydrate Kinases

GHMP carbohydrate kinases share the general structure of their wider fold, with two α/β domains of approximately equal size (Figure 5a). The two domains are not strictly separated in the protein sequence, with the very C-terminus of the protein contributing to the domain that is largely made from the *N*-terminal parts of the protein [99,100]. Some members of the family are monomeric, whilst others are obligate dimers [36,87]. The protein active site is formed in a cleft between the two domains. The carbohydrate substrate is held in a deep pocket that is formed between the two domains, whilst the nucleotide binds along a shallow groove between the two domains with parts of the ribose ring exposed to solvent [87]. The nucleotide binding site is present before the carbohydrate binds to the enzyme, similarly to the RKs. This permits different GHMP kinases to show a variety of kinetic mechanisms. Eukaryotic galactokinases show an ordered mechanism, where ATP binds the enzyme before galactose [100–102], as do *N*-acetylgalactosamine kinases [92]. Plant galactokinases and bacterial heptose-7-phosphate kinases show the opposite mechanism, an ordered mechanism with the carbohydrate binding first [87,103]; whilst, *E. coli* galactokinase shows a random sequential mechanism [104].

Two amino acid side chains are strongly conserved throughout the GHMP carbohydrate kinases (aspartic acid and arginine; Figure 5b) [99,100,105]. The aspartic acid coordinates the carbohydrate substrate 1′ hydroxyl group. This has been proposed to act as a catalytic base [106], deprotonating the reacting 1′ hydroxyl group to make it a stronger nucleophile to attack the γ-phosphate group of ATP [87,107]. The arginine coordinates the sugar and the ATP phosphate groups. It is expected to assist in increasing the $pK_a$ of the conserved aspartic acid to make this a better catalytic base [106,107]. Mutation of the conserved aspartate abolished activity in mammalian galactokinases [16,106–108] and heptose-7-phosphate kinase [87]. Conservative mutation of the conserved arginine showed a small reduction in catalytic efficiency [107]. Non-conservative mutations abolish activity [87,107], but they may also reduce enzyme stability. A conserved sequence of three serine residues helps to coordinate a divalent cation (usually magnesium) and the ATP phosphates [87]. The pentacoordinate transition in the kinase reaction (Figure 2c) is stabilized by the cation, the conserved arginine, and an anion hole formed from backbone amides [87,107]. One quantum mechanics/molecular mechanics (QM/MM) study supported the catalytic base mechanism [107], with the catalytic aspartate polarizing the O-H bond in the reactive hydroxyl to facilitate catalysis. Another study instead proposed that the reaction proceeds by direct attack of the hydroxyl oxygen on ATP, with the intermediate stabilized by a pair of arginine residues [109]. A similar mechanism was proposed for another GHMP kinase [110]. As the critical proposed catalytic arginine is not conserved [109], and non-conservative mutations have little effect on catalytic efficiency [105], it is more probable that the true mechanism is a combination of those proposed (i.e., a catalytic base mechanism assisted by the stabilization of negative charge on the ATP phosphates). This remains controversial and would benefit from similar detailed studies on a wider range of GHMP carbohydrate kinases. The catalytic base mechanism is similar to that observed in the hexokinases, with the aspartic acid and magnesium playing identical roles; whilst, the anion hole is

similar to that seen in ribokinases. Here, the conserved arginine provides additional catalytic power, which may reflect the less reactive nature of the anomeric oxygen in comparison to the 6′ oxygen.

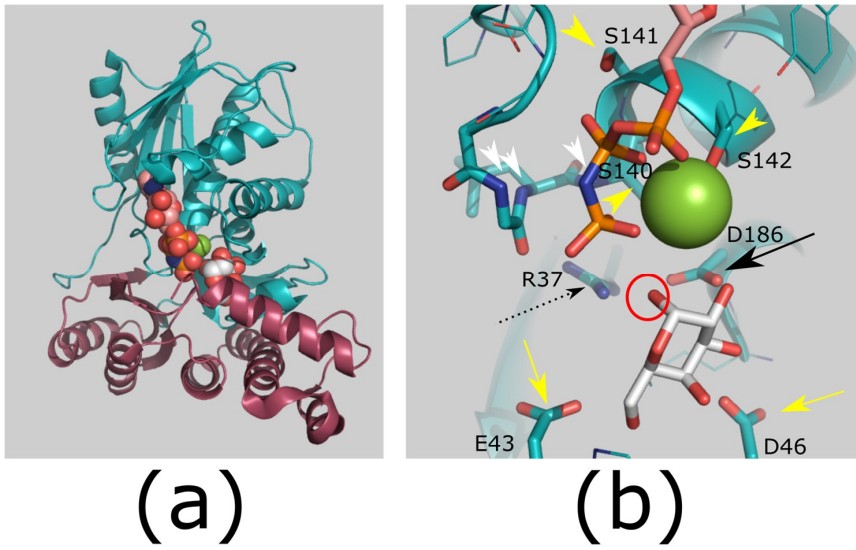

**Figure 5.** Structure and mechanism of GHMP kinases. (**a**) Overview of GHMP kinase structure. The structure forms two domains of approximately equal size (shown in teal and raspberry; PDB ID: 1WUU [36]). (**b**) Proposed mechanism for GHMP kinases. The reactive hydroxyl (red circle) is deprotonated by an aspartic acid (catalytic base; black arrow; D186). This will attack the γ-phosphate of ATP. The aspartic acid $pK_a$ is increased by an arginine residue (black dashed arrow; R37), which also coordinates the ATP phosphates. The pentacoordinate transition state is stabilized by this arginine side chain, a cation (green sphere), and by an anion hole produced by main chain amides (white arrowheads). Additional side chains help to bind the sugar hydroxyls (yellow arrow; E43/D46), the cation, and the ATP phosphates (yellow arrowheads; S140/S141/S142). Colors: ligand bound enzyme (teal; PDB ID: 1WUU). Ligands galactose (white carbons; substrate) and AMP-PNP (carbons pink; substrate analog). Images generated using PyMOL version 1.8.

## 6. Phosphatidylinositol Related Kinases (PI-Ks)

### 6.1. Overview of the Family and Specificity

Phosphatidylinositol (PtdIns) is a minor phospholipid that is found on the cytosolic leaflet of eukaryotic cellular and organelle membranes. It can be phosphorylated at multiple free hydroxyl groups, apart from the 2′ and 6′ positions. Alternatively phosphorylated PtdIns are used by eukaryotic cells as markers of different organelles and as second messengers [111–114]. Phosphorylation of PtdIns has many important roles in signalling and physiology, with mutations leading to familial diseases [115,116]. This phosphorylation is performed by a distinct class of enzymes, the PtdIns kinases (PI-Ks). These are divided into the PtdIns-3-kinases (PI3Ks), PtdIns-4-kinases (PI4Ks), and phosphatidylinositol phosphate kinases (PIP5Ks) [111]. The PI3K and PIP5K groups are further classified into types I, II, and III, according to their structural features and lipid substrate preferences [111,117,118]. Type I and type II PI5Ks generate phosphatidylinositol 4,5-bisphosphate (also known simply as PIP2 or PI(4,5)P2) from phosphatidylinositol 4-phosphate (PI4P or PIP) and phosphatidylinositol 5-phosphate (PI5P), respectively, whereas the type III produces phosphatidylinositol 3,5-bisphosphate (PI(3,5)P2) from phosphatidylinositol 3-phosphate (PI3P) [119]. Mutations in PI3Ks that enhance activity have a long history of association with cancer [120–122]. A single class III PI3K ortholog is conserved from lower eukaryotes to plants and mammals. This class is mainly responsible for the generation of a large fraction of the PtdIns-3-phosphate, a central phospholipid for membrane trafficking processes [123]. PtdIns kinases appear to strongly prefer ATP as the phosphate donor [119].

## 6.2. Structure and Catalytic Mechanism of PIPKs

The PtdIns kinases are all large, multidomain proteins with substantial regulatory elements. The PI3K catalytic subunits have a three-domain core structure consisting of a C2 domain, a helical domain, and a catalytic domain (Figure 6a) [113]. The 110 kDa class I PI3Ks form heterodimers with a regulatory subunit in the cell [124]. Some isoforms form obligate heterodimers with a single regulatory partner, whilst the p110γ isoform can choose between several distinct regulatory partners [117]. In contrast, the ~170 kDa class II PI3Ks are monomeric, with an additional large *N*-terminal region that lacks homology to other known proteins [114]. Other PtdIns kinases show similar complexes with regulatory subunits [119,125,126].

The molecular basis for PtdIns kinase catalysis and signaling specificity remains controversial. Although structures of ligand-free kinases have been solved, these could not confirm the catalytic residues [127,128]. A recent study by Muftuoglu et al. has suggested a basis for catalysis and specificity in some PtdIns kinases [119]. Crystallographic analysis of the zebrafish type I kinase PIP5Kα revealed that a lysine side chain interacts with the ATP γ-phosphate (Figure 6b). This and an arginine side chain provide a positive charge to stabilize the pentacoordinate transition state. Modeling PI4P into the PIP5Kα active site of is consistent with the arginine and lysine, also providing affinity for the monophosphate on the inositol head group. Manganese ions are coordinated by a network of side chains and they provide further stabilization to the transition state. These side chains are conserved throughout PIP5Ks, suggesting that ATP binding is similar throughout this family [129]. In contrast, the lipid sugar has to be arranged such that the reactive hydroxyl is positioned near to the ATP γ-phosphate. Different substrates have phosphates that are attached to differing positions in the inositol ring. This ring consequently has to flip 180° to project the phosphate of different substrates into the common binding pocket [119]. A recent mutational and kinetic study on PI3Kα phosphorylation of PIP2 provided insights into in the catalytic mechanism of this class [129]. A conserved lysine residue assists in recognition of both ATP and substrate. The proximity of this to the reactive hydroxyl helps to lower the pK$_a$ for this hydroxyl, facilitating proton abstraction. A histidine side chain is conserved in PI3Ks, but it is not found in related protein kinases. This is proposed to be the PI3K catalytic base, which deprotonates the reactive hydroxyl in a similar manner to aspartic acids in other carbohydrate kinases. The reduction in hydroxyl pK$_a$ likely allows the less basic histidine to act in this case (Figure 6c). A second histidine is located close to the ATP γ-phosphate and it likely assists in stabilizing the pentacoordinate transition state. Kinetic analysis of PI3Kα indicated that this enzyme shows an ordered sequential mechanism [129].

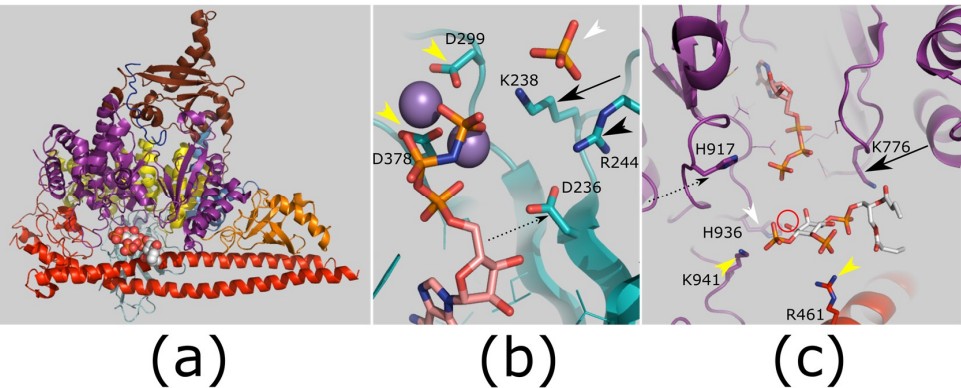

**Figure 6.** Structure and mechanism of PtdIns kinases. (**a**) Overview of PtdIns-3-kinases (PI3K) complex with substrate. PI3K is shown as cartoon, with domains shown in different colors (kinase domain: purple; regulatory subunit: red). Substrate PIP2 shown in spheres (white carbons). PDB ID: 4OVV [130]. (**b**) Proposed mechanism for PIP5K activity. Protein structure is shown in teal; Ligands AMP-PNP (carbons pink; substrate analog), selenate (orange selenium, white arrowhead; analog of sugar phosphate group) and manganese (purple spheres). A conserved lysine (black arrow; K238) and

arginine (black arrowhead; R244) coordinate the substrate phosphate, and are essential for activity. The lysine will also stabilize the pentacoordinate transition state with the manganese ions. An invariant aspartic acid (black dashed arrow; D236) is also required for activity, and it is mutated in human disease: this may contribute to activating the reactive hydroxyl. The manganese ions coordinate to the ATP phosphates and protein side chains (yellow arrowheads; D299/D378). PDB ID: 5E3U [119]. (**c**) Proposed mechanism of PI3Ks. Protein structure colored as (**a**); ATP (carbons pink, substrate) location taken from PDB ID: 1E8X [131]. The reactive hydroxyl (red circle) is proposed to be deprotonated by a histidine side chain (catalytic base; black dashed arrow; H917). This will attack the $\gamma$-phosphate of ATP. A conserved lysine residue (black arrow; K776) coordinates the PIP2. A further conserved histidine (white arrowhead; H936) is essential for reaction. Additional side chains help to bind the substrate phosphates (yellow arrowheads; R461/K941). Images generated using PyMOL version 1.8.

## 7. Perspective and Conclusions

Carbohydrate kinases are attractive targets for therapeutics, with human glucokinase and galactokinase-1 being particularly actively pursued. They belong to several different structural families and catalyze reactions on a wide variety of carbohydrate units. Nevertheless, the different structural classes have all arrived at a very similar catalytic mechanism. In all five cases, the core catalytic mechanism is the deprotonation of the reactive hydroxyl by a catalytic base. This is usually a highly conserved aspartic acid, but, in the case of the PtdIns kinases, a positively charged residue from the conserved sugar phosphate binding pocket increases the sugar hydroxyl acidity allowing for a histidine to act as the base. Following this, the deprotonated oxygen performs a nucleophilic attack on the ATP $\gamma$-phosphate. Each enzyme stabilizes the pentacoordinate transition state with positively charged groups: a divalent cation, and in some cases a conserved positively charged side chain. Despite this strong similarity in the catalytic method, the enzymes show a range of kinetic mechanisms: all are Bi–Bi enzymes with two substrates and two intermediates, but the order in which the substrates must bind differs between enzymes. As many carbohydrate kinases are critical decision points in metabolic pathways, there are clear examples of allosteric regulation from several families. However, even structurally similar enzymes use very different mechanisms of regulation, suggesting that allosteric regulation can evolve independently according to the need of the biological system. Carbohydrate kinases combine elegant simplicity of reaction mechanism and the broad capability for modulation.

**Author Contributions:** Conceptualization, S.R. and N.J.H.; investigation, S.R. and M.V.V.; writing—original draft preparation, S.R.; writing—review and editing, M.V.V. and N.J.H.

**Funding:** This research was funded by the Biotechnology and Biological Sciences Research Council, grant number BB/N001591/1 and the APC was funded by the University of Exeter.

**Acknowledgments:** The authors thank Alice Cross (University of Exeter) for helpful discussions in framing this review.

**Conflicts of Interest:** The authors declare no conflict of interest. The funders had no role in the conception of the review, in the writing of the manuscript, or in the decision to publish the results.

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
