# Peer review of "Carbohydrate Kinases: A Conserved Mechanism Across Differing Folds"

_catalysts, doi:10.3390/catal9010029_

Round 1
Reviewer 1 Report
The authors present a review of the major sugar kinase families. This is an interesting subject for a brief review such as this. However, the authors make the assumption that all these enzymes function through an active site base mechanism. While most papers do say that this is the case, it is not universally accepted. There are some QM/MM studies suggesting that this is not the case, partially backed up by structural and experimental evidence. I think that it would be ok for the authors to discuss these studies and conclude that, on the balance of probabilities, they are likely to incorrect - but they must discuss them. There are also some points in the review here I feel that the authors could provide a more complete discussion of the experimental evidence for the mechanisms described.
Major issues:
In Figure 2c the mechanism suggests that there is a (perhaps short-lived) covalent ADP-P-sugar complex. Has this ever been isolated? What is the evidence for its existence. The authors should note that this is a chemically different mechanism to the one they describe for the GHMP kinases where the C1-OH is deprotonated and this causes the transfer of the phosphate group without a covalent intermediate.
What is the experimental evidence for the mechanism described for the ribokinase family? The authors should briefly describe (and provide references for) any kinetic, structural, mutagenic and computational studies.
Many references appear to be incorrect. This seems to be a mixture of not updating the list (so there are some "frameshift" issues) and also the citing of inappropriate sources. Some examples (but please check every reference) - Line 289: Also needs reference for GALK1 as a target in the treatment of galactosemia (first proposed by Bosch in JIMD). Line 313: Ref 88 does not provide details on the mechanism of mammalian galactokinase (the correct references are a mixture of papers from the 1960s/1970s and 2000s probably cited in Ref 87).
Line 297: These are completely the wrong refs. There are two papers (both published around 2011) suggesting that human GALK2 has much more relaxed specificity than GALK1. These should be cited. The authors should also note that bacterial galactokinases are much less specific that mammalian ones (work from Thorson's lab and others - check the literature).
Ref 93 is not the only paper to study active site mutations in human galactokinase. There is one by Kent Lai's group (in ChemicoBiological Interactions, I think) and another by Hazel Holden's (FEBS Letts). Please check the literature to see if there are more. Ref93 suggests that protein misfolding may be occurring and may explain the loss of activity.
The authors should cover the work from Meilan Huang's group suggesting a different mechanism for galactokinase (and, by inference, other GHMP kinases).
Minor issues:
In the legend to Figure 2, the Greek letter before phosphate is missing.
Adenine kinase is, technically, not a sugar kinase
Author Response
The authors present a review of the major sugar kinase families. This is an interesting subject for a brief review such as this. However, the authors make the assumption that all these enzymes function through an active site base mechanism. While most papers do say that this is the case, it is not universally accepted. There are some QM/MM studies suggesting that this is not the case, partially backed up by structural and experimental evidence. I think that it would be ok for the authors to discuss these studies and conclude that, on the balance of probabilities, they are likely to incorrect - but they must discuss them. There are also some points in the review here I feel that the authors could provide a more complete discussion of the experimental evidence for the mechanisms described.
We are grateful to the referee for a thoughtful and well-considered referee’s report, and for drawing our attention to some literature that we were not aware of. We have amended the review in the light of these excellent comments.
Major issues:
In Figure 2c the mechanism suggests that there is a (perhaps short-lived) covalent ADP-P-sugar complex. Has this ever been isolated? What is the evidence for its existence. The authors should note that this is a chemically different mechanism to the one they describe for the GHMP kinases where the C1-OH is deprotonated and this causes the transfer of the phosphate group without a covalent intermediate.
Referee 2 suggested a different visualisation of this species. We have also discussed this with an experienced inorganic chemist. Following the excellent points from the referees and our colleague, we have modified the mechanism to show a pentacoordinate transition state, rather than a discrete covalent intermediate. This does not affect the associated discussion, but is a better reflection of the chemical reality, as the referees suggest.
What is the experimental evidence for the mechanism described for the ribokinase family? The authors should briefly describe (and provide references for) any kinetic, structural, mutagenic and computational studies.
We have added further references to highlight the mechanism for the ribokinase family as suggested (the review as a whole has 17 references added).
Many references appear to be incorrect. This seems to be a mixture of not updating the list (so there are some "frameshift" issues) and also the citing of inappropriate sources. Some examples (but please check every reference) - Line 289: Also needs reference for GALK1 as a target in the treatment of galactosemia (first proposed by Bosch in JIMD). Line 313: Ref 88 does not provide details on the mechanism of mammalian galactokinase (the correct references are a mixture of papers from the 1960s/1970s and 2000s probably cited in Ref 87).
We thank the referee for pointing these issues out. We have checked all the references for the revision, and have amended the two points that the referee noted specifically.
Line 297ff: These are completely the wrong refs. There are two papers (both published around 2011) suggesting that human GALK2 has much more relaxed specificity than GALK1. These should be cited. The authors should also note that bacterial galactokinases are much less specific that mammalian ones (work from Thorson's lab and others - check the literature).
We have amended the manuscript in the light of this comment to note that some GHMP kinases have relaxed specificity, and added extra references.
Ref 93 is not the only paper to study active site mutations in human galactokinase. There is one by Kent Lai's group (in ChemicoBiological Interactions, I think) and another by Hazel Holden's (FEBS Letts). Please check the literature to see if there are more. Ref93 suggests that protein misfolding may be occurring and may explain the loss of activity.
We have added reference to these two additional papers (and an extra relevant reference), and included the details of these in the review.
The authors should cover the work from Meilan Huang's group suggesting a different mechanism for galactokinase (and, by inference, other GHMP kinases).
We have referenced the work of the Huang group, and discussed the alternative mechanism proposed. We add that the mechanism remains controversial and would be an excellent area for future research.
Minor issues:
In the legend to Figure 2, the Greek letter before phosphate is missing.
Adenine kinase is, technically, not a sugar kinase
We have amended the manuscript to address these minor issues.
Reviewer 2 Report
The authors have submitted a review on carbohydrate kinases, which is a topical and interesting are of enzyme research, with some potential for drug design and maybe application. On the whole, the writing style is clear, and the review is well illustrated (except for labels)and features appropriate citations.
In the Abstract and throughout- the term ‘pentahedral’ (five faces) seems wrong – I think the authors mean ‘pentacoordinate’ (five partial bonds); also I think they mean ‘transition state’ (TS) – ‘intermediate’ means something very different, and there has been an enormous amount of controversy about the ‘isolation’ of pentavalent phosphate ‘intermediates’ in the literature in the last years, so the authors should check very carefully what they mean. For example, ref 17, cited by the authors, actually states that... ‘The resulting transition state would have a pentacovalent geometry..’ and there are sure to be other examples of paraphrasing within the review. These should be checked.
Other, minor comments:
Page 4 line 101-102 – rephrase ‘…class enzymes from many eukaryotes and prokaryotes have been reported.
Page 4 line 117 – again check use of ‘intermediate’
Page 5 Figure 2 – need to sort out the clashing atoms in the carbohydrate structures
Page 5 figure 2 legend – check ‘pentahedral’/’intermediate’
Page 5 figure 2 – two of the TS bonds should be partial? – also cannot show movement of 2 electrons within a TS thus?
Page 5 Figure 2, 3, 4, 5 etc. need amino acid side chain labels in the active site diagrams if they are to be clear
Page 5 line 144 and throughout – italicise heteroatoms
Page 6 line 157 and throughout - italicise organism names; also page7 line 207, page 8 line 250 etc. etc.
Page 6 line 174-175 - check ‘pentahedral’/’intermediate’
Figure 9 line 273 - check ‘pentahedral’/’intermediate’
Page 11 line 374 ‘et al.’ – also remove ‘Y’
Page 13 –the review ends rather abruptly with reference to the enzymes being ‘attractive targets for drug design’ – it would be good if a short summary of the contribution of the structure/mechanism studies described to the successful design of drugs could be added, either at appropriate sections within the review or at the end.
Author Response
The authors have submitted a review on carbohydrate kinases, which is a topical and interesting are of enzyme research, with some potential for drug design and maybe application. On the whole, the writing style is clear, and the review is well illustrated (except for labels)and features appropriate citations.
In the Abstract and throughout- the term ‘pentahedral’ (five faces) seems wrong – I think the authors mean ‘pentacoordinate’ (five partial bonds); also I think they mean ‘transition state’ (TS) – ‘intermediate’ means something very different, and there has been an enormous amount of controversy about the ‘isolation’ of pentavalent phosphate ‘intermediates’ in the literature in the last years, so the authors should check very carefully what they mean. For example, ref 17, cited by the authors, actually states that... ‘The resulting transition state would have a pentacovalent geometry..’ and there are sure to be other examples of paraphrasing within the review. These should be checked.
We completely agree with the referee’s point that it would be better to refer to a pentacoordinate transition state rather than an intermediate for clarity. We have made this change throughout the manuscript and ensured that the remainder of the discussion is consistent with this.
Other, minor comments:
Page 4 line 101-102 – rephrase ‘…class enzymes from many eukaryotes and prokaryotes have been reported.
Page 4 line 117 – again check use of ‘intermediate’
Page 5 Figure 2 – need to sort out the clashing atoms in the carbohydrate structures
Page 5 figure 2 legend – check ‘pentahedral’/’intermediate’
Page 5 figure 2 – two of the TS bonds should be partial? – also cannot show movement of 2 electrons within a TS thus?
Page 5 Figure 2, 3, 4, 5 etc. need amino acid side chain labels in the active site diagrams if they are to be clear
Page 5 line 144 and throughout – italicise heteroatoms
Page 6 line 157 and throughout - italicise organism names; also page7 line 207, page 8 line 250 etc. etc.
Page 6 line 174-175 - check ‘pentahedral’/’intermediate’
Figure 9 line 273 - check ‘pentahedral’/’intermediate’
Page 11 line 374 ‘et al.’ – also remove ‘Y’
Page 13 –the review ends rather abruptly with reference to the enzymes being ‘attractive targets for drug design’ – it would be good if a short summary of the contribution of the structure/mechanism studies described to the successful design of drugs could be added, either at appropriate sections within the review or at the end.
We thank the referee for carefully reading through the manuscript and noting these minor issues. We have amended all of them as suggested, in particular amending all the figures and legends that the referee pointed out. We have amended the ending of the review to finish more strongly; and have highlighted key therapeutic targets in the main text.
Round 2
Reviewer 1 Report
Thank you for addressing my queries. I recommend this article for publication.
Reviewer 2 Report
The suggested corrections have been made.